# Hidden Dissipation and Irreversibility in Maxwell’s Demon

**DOI:** 10.3390/e24010093

**Published:** 2022-01-06

**Authors:** Paul W. Fontana

**Affiliations:** Physics Department, Seattle University, 901 12th Ave., Seattle, WA 98122, USA; fontanap@seattleu.edu; Tel.: +1-206-296-5934

**Keywords:** Maxwell’s demon, nonequilibrium thermodynamics, irreversibility

## Abstract

Maxwell’s demon is an entity in a 150-year-old thought experiment that paradoxically appears to violate the second law of thermodynamics by reducing entropy without doing work. It has increasingly practical implications as advances in nanomachinery produce devices that push the thermodynamic limits imposed by the second law. A well-known explanation claiming that information erasure restores second law compliance fails to resolve the paradox because it assumes the second law *a priori*, and does not predict irreversibility. Instead, a purely mechanical resolution that does not require information theory is presented. The transport fluxes of mass, momentum, and energy involved in the demon’s operation are analyzed and show that they imply “hidden” external work and dissipation. Computing the dissipation leads to a new lower bound on entropy production by the demon. It is strictly positive in all nontrivial cases, providing a more stringent limit than the second law and implying intrinsic thermodynamic irreversibility. The thermodynamic irreversibility is linked with mechanical irreversibility resulting from the spatial asymmetry of the demon’s speed selection criteria, indicating one mechanism by which macroscopic irreversibility may emerge from microscopic dynamics.

## 1. Introduction

“Maxwell’s demon” refers to a class of thought experiments that probe the universality of the second law of thermodynamics. James Clerk Maxwell published the idea 150 years ago, though descriptions appear earlier in his private communications ([1], p. 308) [2,3]. The demon in its original form sorted molecules of a gas according to their speed to produce a temperature difference between the two sides of a vessel. More generally, the demon is an agent or device that moves a system to a lower-entropy state, generating free energy. According to the second law, the demon must pay the cost for that free energy by generating entropy in itself or the environment so that the net change of entropy of the universe is positive. However, the demon’s sorting mechanism is passive—the original gas demon merely allows molecules to pass from one side of a box to the other if they satisfy some selection criterion, but blocks their passage in the reverse direction—hence there is no obvious means of external entropy generation available, and the demon appears paradoxically to be able to circumvent the second law and generate free energy without doing work [4,5].

For most of the century and a half since Maxwell first proposed the thought experiment an entity that could sort molecules on the molecular scale was beyond the reach of technology, and Maxwell’s demon stood primarily as an abstract conceptual tool for understanding the statistical foundations of thermodynamics. In recent years, though, advances in nanotechnology have brought molecular machines into the realm of possibility, and devices approximating the function of Maxwell’s demon have now been made [6,7,8,9,10,11,12,13,14,15,16,17,18,19,20,21,22,23,24,25,26,27,28,29,30]. Biochemical analogs of Maxwell’s demon are under development for molecular synthesis [31,32]. Miniaturization of computer technology is also approaching the molecular scale, and further advances in computing density and efficiency are constrained by limits imposed by the second law [33,34]. To optimize nanoscale devices and understand their limitations, it is therefore of increasing practical as well as theoretical urgency to resolve the paradox, to ascertain the true thermodynamic cost of molecular sorting imposed by the second law, and to understand the physical constraints that determine that cost.

A well-known resolution from information theory, due to Bennett, invokes Landauer’s principle, which asserts that erasing each bit of physically-encoded information produces a minimum of kln2 of entropy, where *k* is Boltzmann’s constant [35,36,37,38]. It is commonly stated that the entropy production predicted by this principle shows that Maxwell’s demon obeys the second law. However, as others have pointed out, Landauer takes the second law as an assumption and derives the principle as a consequence of it [39,40,41,42,43,44,45,46,47]. Therefore, it only applies to the demon if the demon already obeys the second law, and it does not resolve the paradox. Furthermore, the demon’s action is physically irreversible, so we would expect a true resolution to the paradox to predict a positive value for the net entropy production of the universe during its operation. In fact, the minimum net entropy generation through erasure predicted by Landauer’s principle is zero, just enough to ensure that the entropy of the universe does not decrease and consistent with reversible operation of the demon. Even if Landauer’s principle applies, it cannot account for all of the entropy generation by the demon.

In this paper, the paradox is resolved by a more concrete route that does not depend on information theory or Landauer’s principle, does not assume the demon obeys the second law *a priori*, and yields a quantitative lower bound on entropy production by the demon. The bound is positive in all non-trivial cases, showing that the demon complies with the second law and correctly reflecting the demon’s irreversibility. The key insight is to consider the momentum transport induced by the demon in addition to its associated particle and energy transport. Opening the gate and allowing particles to pass in one direction produces a momentum flux, generating a reaction force on the containing vessel that must be balanced by external forces. Those forces do work on the transmitted particle stream, some of which is necessarily dissipated to the external environment as heat.

Maxwell’s assumption, then, that no work is done on the gas during the demon’s operation is not well taken. The sorting mechanism itself may be a passive gate, but if “the demon” is understood to include all means of external interaction with the gas, including both the sorting mechanism and container, then the operation of the demon necessarily involves external work and heating, even if the energy of the gas itself remains constant. We will compute the external work and heat dissipation, which will lead us to the lower bound on entropy production. We shall see that the bound is positive, providing for the first time a quantitative, logically-consistent resolution to the paradox and reflecting the thermodynamic irreversibility of the demon. Finally, we will briefly examine the mechanical source of the irreversibility and discuss its implications for the thermodynamic arrow of time.

## 2. Analysis

### 2.1. Hidden Dissipation and Rate of Entropy Production in the Environment

Consider a Maxwell’s demon operating on a monatomic ideal gas separated into two volumes. The volumes are isolated from each other, except that the demon, through some unspecified mechanism, permits molecules to exit one gas, the “emitter,” and enter the other, the “receiver,” according to a prescribed rule on their speed, without changing their energy or momentum. We will consider for simplicity a one-way demon that admits particles from the emitter to the receiver, but not the reverse, though the argument is easily generalized to allow for bidirectional flow. Variables are distinguished with a prime for the receiver. For conceptual clarity, we imagine that the two volumes are connected by a conduit or tunnel through which the molecules pass unimpeded after leaving the emitter and before entering the receiver. However, only the apertures of the conduit on the emitting and receiving ends play a role in the dynamics of the system; the conduit is merely a device for separating the action of the demon on the emitter from its action at the receiver, and its length may be taken as zero without any alteration of the analysis, see Figure 1a.

The emitter and receiver are each contained in a box of volume *V*, with *N* or $N^{\prime }$ molecules of mass *m* at temperature *T* or $T^{\prime }$. The conduit connecting the boxes has cross-sectional area *A*. The *z*-axis aligns with the conduit, with positive *z* towards the receiver. The stream of particles passing from the emitter to the receiver through the conduit constitutes a beam in the +z direction.

Let us take the following assumptions as premises, chosen so that our model will reflect Maxwell’s original thought experiment as closely as possible:The “system” includes only the gas in the two chambers, since it is the state of the gas alone whose change we seek to reconcile with the second law. In particular, the filtering or sorting mechanism is not considered to be part of the system. The sorting mechanism is assumed to be passive in the sense that it does not change the energy or momentum of particles that it allows to pass from one chamber to the other, while particles that it blocks are reflected elastically as from a perfectly rigid wall. This is in the spirit of the original thought experiment, in which Maxwell was concerned only with the temperature difference and deliberately designed his demon not to alter the state of the particles it chose to let pass.The total internal energy of the gas does not change during the demon’s operation. This assumption is implicit in Maxwell’s original statement of the thought experiment, where he describes the gas as “enclosed in an envelope which permits neither change of volume nor passage of heat,” while the demon operates “without expenditure of work.” ([1], p. 308). Our assumption is weaker, since Maxwell assumed separately that the work done on the gas and the heat transferred to it are individually zero. In what follows, however, we will show that the assumption that no work is done on the gas is not consistent with Newton’s laws as applied to the transport of momentum from one chamber to the other during the demon’s operation. We will therefore not assume that work and heat are separately zero, but rather that any work done on the system by external forces is dissipated as heat in the environment so that the net effect is zero change to the internal energy of the system.The chambers holding the gas are held rigidly in place by some mounting structure exerting external forces on the walls, and therefore on the gas. The chamber walls are the system-facing components of the environment and can in principle do work on and exchange heat with the system. However, in light of premise 2, any work done by these external forces must be dissipated as heat in the environment. In the analysis of forces below, the sorting mechanism and chamber walls are collectively referred to as “the container,” since it is unnecessary and less general to distinguish whether forces on the gas are exerted by the sorting device or the chamber walls.The walls of each chamber are in thermal contact with a thermal reservoir into which the heat generated by the demon’s operation is dissipated. These reservoirs are “natural” thermal systems in the sense that they obey the second law and have a well-defined temperature, so that heat dQ deposited into them increases their entropy by at least dQ/T. They are not capable, for instance, of converting the heat they absorb entirely into work. This may appear to be a back-door assumption that the demon obeys the second law, but it is not. We are only assuming that some portion of the environment receiving heat from the operation of the demon is itself demon-free. In the spirit of the original thought experiment, we will assume the demon’s only direct actions are on the system, not on the environment, and in particular not on the functioning of the thermal reservoirs. The consequent entropy gain of these reservoirs will be seen to compensate for the decrease in entropy of the system.

The first law of thermodynamics for the emitter and receiver may be written as follows: (1)$$\begin{array}{ccc} \dot{U} & = & -\dot{K}+{\dot{W}}_{mech}+{\dot{W}}_{chem}-{\dot{Q}}_{diss} \end{array}$$(2)$$\begin{array}{ccc} {\dot{U}}^{\prime } & = & +\dot{K}+{\dot{W}}_{mech}^{\prime }+{\dot{W}}_{chem}^{\prime }-{\dot{Q}}_{diss}^{\prime }, \end{array}$$
where $\dot{U}$ is the rate of change of the internal energy of the gas, $\dot{K}$ is the rate of gain of kinetic energy of the beam, that is, the rate of energy transport out of the emitter and into the receiver, ${\dot{W}}_{mech}$ is the rate of doing mechanical work on the subsystem by the external force, that is, by the demon, ${\dot{W}}_{chem}$ is the rate of doing chemical work associated with temperature (discussed below), ${\dot{Q}}_{diss}$ is the rate of external heat dissipation, and primes indicate the corresponding quantities at the receiver (Figure 2). The $\pm \dot{K}$ terms can equivalently be interpreted as the rate of doing mechanical work on the beam by the gas at the emitting and receiving ends, respectively. In light of premise 2, the sum of (Equation 1) and (Equation 2) is zero, and we may write:(3)$$\dot{U}+{\dot{U}}^{\prime }=0=\left[{\dot{W}}_{mech}-{\dot{Q}}_{mech}\right]+\left[{\dot{W}}_{mech}^{\prime }-{\dot{Q}}_{mech}^{\prime }\right]+\left[{\dot{W}}_{chem}+{\dot{W}}_{chem}^{\prime }-{\dot{Q}}_{chem}-{\dot{Q}}_{chem}^{\prime }\right],$$
where for bookkeeping we have separated ${\dot{Q}}_{diss}$ and ${\dot{Q}}_{diss}^{\prime }$ into parts corresponding to the mechanical and chemical work $\left({\dot{Q}}_{diss}={\dot{Q}}_{mech}+{\dot{Q}}_{chem},\;{\dot{Q}}_{diss}^{\prime }={\dot{Q}}_{mech}^{\prime }+{\dot{Q}}_{chem}^{\prime }\right)$ and grouped the terms according to processes involving mechanical work on the emitter, mechanical work on the receiver, and chemical work on the system. We will assume that each of these groups is independently zero, allowing us to calculate dissipation to the environment associated with each process. This is merely a matter of notation to give a quantitative representation of premise (2) and is not an additional assumption on the physics.

Let the distribution function f(v) be defined such that the fraction of particles with velocities in a velocity-space volume $d^{3}v$ about v is $f\left(v\right)d^{3}v$. The average of a quantity *x* over the distribution is $\langle x\rangle =\int_{admissible}xf\left(v\right)\;d^{3}v$, where the average will be taken over the part of the distribution with admissible particles according to whatever selection rule the demon applies.

The number of particles per unit time with velocity in the neighborhood of v incident on the aperture is the number density of such particles: $\frac{Nf\left(v\right)d^{3}v}{V}$, times the volume they sweep out in time dt: $Av_{z}dt$, divided by the time interval dt:$$d\dot{N}=\frac{NA}{V}v_{z}f\left(v\right)d^{3}v,$$
so that the total number of particles leaving the emitter and entering the beam per unit time is
(4)$$\dot{N}=\frac{NA}{V}\langle v_{z}\rangle .$$

Similarly, the z-momentum incident per unit time for particles in the neighborhood of v is $d{\dot{J}}_{z}=\left(mv_{z}\right)d\dot{N}=\frac{NA}{V}mv_{z}^{2}f\left(v\right)d^{3}v$, so the rate of change of momentum of the beam is
(5)$${\dot{J}}_{z}=\frac{NA}{V}m\langle v_{z}^{2}\rangle .$$

The differential incident kinetic energy per unit time is $d\dot{K}=\left(\frac{1}{2}mv^{2}\right)d\dot{N}=\frac{NA}{V}\frac{m}{2}v^{2}v_{z}f\left(v\right)d^{3}v$, so energy enters the beam at the rate
(6)$$\dot{K}=\frac{NA}{V}\frac{m}{2}\langle v_{⊥}^{2}v_{z}+v_{z}^{3}\rangle ,$$
where $v_{⊥}^{2}=v_{x}^{2}+v_{y}^{2}$.

By Newton’s second law, the rate of change of momentum $d{\dot{J}}_{z}$ is the net force on beam particles with velocity in the neighborhood of v. The rate of doing mechanical work by a force F on a particle with velocity v is F·v, so the net rate of doing mechanical work on beam particles as they enter the beam is $d\dot{W}=\left(d{\dot{J}}_{z}\right)v_{z}$, and the rate of doing work on the whole distribution of beam particles is:(7)$$\dot{W}=\frac{NA}{V}m\langle v_{z}^{3}\rangle .$$

When operating, the transmission of particles will deplete the emitter’s distribution function and augment the receiver’s in the range of admitted velocities, so that in steady state the distributions will not in general be Maxwellian. In what follows, we will restrict our attention to the case where the beam intensity is low enough that the distortion to the emitter’s and receiver’s distribution functions is small, that is, the gases thermalize quickly relative to the rate of particle loss. If τ is the greater of the thermalization times of the emitter and receiver, then this condition may be written as $\tau \dot{N}/N<<1$. This regime can always be achieved, for example, by making the aperture *A* sufficiently small. With this assumption, the gases may be considered to be thermal at any given time, and thermal macrovariables, such as gas pressure and temperature are unambiguously defined. While this assumption limits the generality of our result, it allows the result to be expressed in terms of the temperatures of the gases and makes the interpretation of the dissipation mechanism derived below more transparent.

### 2.2. Analysis of Forces and Mechanical Work

We wish first to determine ${\dot{W}}_{mech}$, the portion of $\dot{W}$ done by the demon, that is, by external forces, and what the resulting change in entropy of the environment is. On the emitting side, there are three mutually-interacting agents: The emitting gas, *e*, the beam, *b*, and the container (including the sorting mechanism or “demon” and the walls of the chamber), *c*. Let us consider the forces among these elements, depicted schematically in Figure 1. First, we note that the z-stress on the aperture on the gas side is $P-{\dot{J}}_{z}/A$, where *P* is the isotropic pressure of the emitter. This stress is reduced from the value *P* it would have if the aperture were closed, because beam particles that are incident on the wall are not reflected by it and so receive no impulse from it. On the beam side the z-stress is ${\dot{J}}_{z}/A$. Hence the force on the beam by the emitter is
(8)$$F^{b,e}=PA-{\dot{J}}_{z}.$$

Since by Newton’s second law the net force on the beam is $F^{b}={\dot{J}}_{z}$, the force on the beam by the container is
(9)$$F^{b,c}=F^{b}-F^{b,e}=2{\dot{J}}_{z}-PA,$$
see Figure 1b.

The forces on the gas and container shown in Figure 1d,f do no mechanical work because they act on elements at rest, but we include them for completeness. As the emitting gas gains no net momentum during the operation of the demon, the net force $F^{e}$ on it is zero, while Newton’s third law requires that there be a reaction force on the gas from the beam, $F^{e,b}=-PA+{\dot{J}}_{z}$. The force from the container must cancel this, i.e.,
(10)$$F^{e,c}=PA-{\dot{J}}_{z},$$
see Figure 1d. Forces $F^{c,e}$ and $F^{c,b}$ in Figure 1f are reactions to the forces on the gas and the beam by the container, $F^{e,c}$ and $F^{b,c}$ respectively. We see from Newton’s second law that the external support structure must exert a force $F^{c,ext}={\dot{J}}_{z}$ on the emitting chamber to keep it in place.

The container does mechanical work on the beam via the force (Equation 9). The differential rate of doing work by this force is $d{\dot{W}}_{mech}=\left(dF^{b,c}\right)v_{z}$, so the rate of doing mechanical work on the system at the emitter end is
(11)$${\dot{W}}_{mech}=\frac{2NAm}{V}\langle v_{z}^{3}\rangle -PA\langle v_{z}\rangle =2\dot{W}-k\dot{N}T,$$
where we have used the ideal gas law PV=NkT, (Equation 4), and (Equation 7) to go from the second expression to the third. In the quasi-thermal regime considered here, the distribution function remains Maxwellian, for which it can be shown that for any prescribed speed filter, $\langle v_{⊥}^{2}v_{z}\rangle =\langle v_{z}^{3}\rangle$ (see Appendix A). With (Equation 6) and (Equation 7) this implies:(12)$$\dot{W}=\dot{K},$$
so that the first term of (Equation 11) represents twice the rate of change of kinetic energy of the beam. The second term is a reduction in the external work required to release the beam because internal work is done on the beam by the emitter gas itself. According to the assumptions associated with Premise 2 and Equation (Equation 3), the work (Equation 11) must be entirely dissipated as heat externally: ${\dot{Q}}_{mech}={\dot{W}}_{mech}$, Figure 2b. If this heat is dissipated into a thermal reservoir according to premise 4, the minimum entropy production occurs if the heat is dissipated reversibly and the bath is at the same temperature *T* as the emitter, so the rate of increase of the entropy of the environment is bounded by
(13)$${\dot{S}}_{mech}\geq \frac{{\dot{Q}}_{mech}}{T}=\frac{2\dot{K}}{T}-k\dot{N}.$$

At the receiving end, the stress on the gas side of the aperture is $P^{\prime }+{\dot{J}}_{z}/A$, and on the beam side is $-{\dot{J}}_{z}/A$. The forces exerted by the container are (see Figure 1c,e).
(14)$$\begin{array}{ccc} F^{\prime b,c} & = & P^{\prime }A \end{array}$$
(15)$$\begin{array}{ccc} F^{\prime r,c} & = & -\left(P^{\prime }A+{\dot{J}}_{z}\right). \end{array}$$

As on the emitter side, force (Equation 15) and the forces on the gas and the container shown in Figure 1e,g do no work because they are acting on stationary elements. It is interesting to note that although the activity of the demon is apparently entirely internal to the chamber and the net external force $F^{c,ext}+F^{\prime c,ext}$ is zero, an external observer would nevertheless be able to detect when the demon was in operation by the added pressure trying to force the emitter and receiver apart. To keep the two halves of the box together during the demon’s operation, an inward force couple of magnitude ${\dot{J}}_{z}$ must be applied (perhaps by the material joining the two halves of the box).

The container does mechanical work on the beam at the receiver through the force (Equation 14). The rate of doing mechanical work by the container at the receiver is thus
(16)$${\dot{W}}_{mech}^{\prime }=P^{\prime }A\langle v_{z}\rangle =k\dot{N}\frac{N^{\prime }T^{\prime }}{N}.$$

Assuming that this work is all dissipated as heat externally into a heat bath at temperature $T^{\prime }$, we have ${\dot{Q}}_{mech}^{\prime }={\dot{W}}_{mech}^{\prime }$ (see Figure 2a), and the rate of change of the entropy of the environment is bounded by
(17)$${\dot{S}}_{mech}^{\prime }\geq \frac{{\dot{Q}}_{mech}^{\prime }}{T^{\prime }}=k\dot{N}\frac{N^{\prime }}{N}.$$

### 2.3. Analysis of Chemical Work

In addition to mechanical work, we must also consider chemical work. This is the work associated with isothermal changes in the macrovariables *N* and $N^{\prime }$. If a subsystem at temperature *T* gains dN particles with no energy, its internal energy must decrease by $\frac{3}{2}kTdN$ to thermalize them. Hence the variable thermodynamically conjugate to *N* is $-\frac{3}{2}kT$, and the chemical work done on the gas is $-\frac{3}{2}kTdN$. (Let us call the quantity $-\frac{3}{2}kT$ the “thermal chemical potential” because it is the chemical potential of a “species” of monatomic ideal gas at temperature *T*. That is, $\frac{3}{2}kT$ is the work required to “create a temperature-*T* particle” from a particle at zero temperature.) In our case, the emitter is losing particles while the receiver is gaining them, so ${\dot{N}}_{gas}=-\dot{N}$, ${\dot{N}}_{gas}^{\prime }=+\dot{N}$, and the net rate of doing chemical work in transferring particles from temperature *T* to temperature $T^{\prime }$ is
(18)$${\dot{W}}_{chem}+{\dot{W}}_{chem}^{\prime }=\frac{3}{2}k\left(T-T^{\prime }\right)\dot{N}.$$

This work is associated with the rethermalization of the transported particles at their destination. For example, if $T>T^{\prime }$ and dN particles are drawn from the emitter at temperature *T*, when they arrive at the receiver they are rethermalized to temperature $T^{\prime }$, releasing energy $\frac{3}{2}k\left(T-T^{\prime }\right)dN$ there. Physically, this energy is then available in the receiving box, and if the system energy is conserved it will be dissipated to the receiver bath at temperature $T^{\prime }$, Figure 2c. Hence the rate of entropy change of the environment due to chemical work is bounded by:(19)$${\dot{S}}_{chem}^{\prime }\geq \frac{{\dot{Q}}_{chem}+{\dot{Q}}_{chem}^{\prime }}{T^{\prime }}=\frac{3}{2}k\dot{N}\left(\frac{T}{T^{\prime }}-1\right).$$

A schematic representation of the energy flow in the system is shown in Figure 2.

The minimum rate of entropy production in the environment due to the operation of the demon is the sum of the entropy produced by mechanical and chemical processes (Equation 13), (Equation 17), and (Equation 19):(20)$${\dot{S}}_{env}={\dot{S}}_{mech}+{\dot{S}}_{mech}^{\prime }+{\dot{S}}_{chem}^{\prime }\geq \frac{2\dot{K}}{T}+k\dot{N}\left[\frac{N^{\prime }}{N}-1+\frac{3}{2}\left(\frac{T}{T^{\prime }}-1\right)\right].$$

### 2.4. Rate of Entropy Production of the System and the Universe

We now show that this entropy production in the environment is sufficient to compensate for the entropy reduction of the system. To find that, we use the Sackur–Tetrode equation for the entropy of a monatomic ideal gas at equilibrium:(21)$$S_{gas}=kN\left(-\ln N+\frac{3}{2}\ln T+constants\right).$$

Taking the time derivative and applying this to both gases gives the net rate of change of entropy of the system as it changes state quasi-statically (see Appendix B):(22)$${\dot{S}}_{sys}={\dot{S}}_{gas}+{\dot{S}}_{gas}^{\prime }=-\dot{K}\left(\frac{1}{T}-\frac{1}{T^{\prime }}\right)-k\dot{N}\ln\frac{N^{\prime }}{N}+\frac{3}{2}k\dot{N}\ln\frac{T^{\prime }}{T}.$$

We recognize the first term as the entropy change associated with reversible heat flow from temperature *T* to temperature $T^{\prime }$. The other two terms, proportional to $k\dot{N}$, represent the entropy change associated with the transfer of particles from the emitter to the receiver in the presence of a difference in number density and in temperature, respectively. As expected, for equal temperatures, entropy increases if the emitter has a higher density than the receiver ($N>N^{\prime }$), consistent with Fick’s law of diffusion. Less obviously, according to the last term, for equal densities and neglecting energy transport, entropy increases if particles are transferred from a low temperature emitter to a higher temperature receiver ($T^{\prime }>T$). This reflects the fact that particles that are extracted from a low temperature environment and introduced into a high temperature environment enjoy more degrees of freedom.

The lower bound on the net rate of change of entropy of the universe is the sum of (Equation 20) and (Equation 22). We group the terms as follows:(23)$${\dot{S}}_{univ}\geq \dot{K}\left(\frac{1}{T}+\frac{1}{T^{\prime }}\right)+k\dot{N}\left(\frac{N^{\prime }}{N}-1-\ln\frac{N^{\prime }}{N}\right)+\frac{3}{2}k\dot{N}\left(\frac{T}{T^{\prime }}-1-\ln\frac{T}{T^{\prime }}\right).$$

The first term is positive if there is net transport of energy. Physically, it represents the entropy increase in the environment due to the hidden dissipation associated with the generation of the momentum in the particle beam at temperature *T* combined with the entropy change of the system due to heat transport from temperature *T* to temperature $T^{\prime }$. The second and third terms are nonnegative because x−1≥lnx for x≥0, with equality only when x=1. The second term can be interpreted to mean that while spontaneous diffusion of particles down a density gradient increases the entropy of the system if $N>N^{\prime }$, doing work to push particles up a density gradient, that is, against the number density-related chemical potential gradient, if $N<N^{\prime }$, is a dissipative action and increases entropy of the environment. The third term can be interpreted to mean that isothermal transfer of particles from low temperature to high temperature increases the entropy of the system if $T<T^{\prime }$, while if $T>T^{\prime }$ the heat released in forcing the particles to move from a high temperature to low against the thermal chemical potential gradient increases the entropy of the environment. In each case, the transfer of particles with or against the corresponding chemical potential gradient increases the entropy of the universe.

The expression (Equation 23) is thus never negative, and Maxwell’s demon obeys the second law of thermodynamics.

### 2.5. Comparison with Landauer’s Principle

We have arrived at our bound (Equation 23) without reference to information theory. Let us now consider the process through the lens of Landauer’s principle [35,36]. Landauer’s argument may be summarized as follows: The erasure of one bit of information corresponds to the compression of an independent degree of freedom from a state with two possible and equally probable microstates to a state with one. This reduces the total number of microstates corresponding to the macrostate by a factor of 2, contracting the phase space of the system to half its volume. By Boltzmann’s law, this decreases the entropy of the system by $\Delta S_{sys}=-k\ln2$. *In order to satisfy the second law of thermodynamics*, there must be enough dissipation in the process to increase the entropy of the environment by at least the same amount, $\Delta S_{env}\geq k\ln2$, *because*
$\Delta S_{univ}$ must be greater than or equal to zero. If the system is in a thermal bath at temperature *T*, then $\Delta S_{env}\geq Q/T$ leads directly to the famous Landauer limit, $Q_{1bit}\geq kT\ln2$. In Landauer’s words, “The entropy of a closed system … cannot decrease; hence this entropy [lost due to erasure] must appear elsewhere as a heating effect, supplying 0.6931 kT per restored [i.e., erased] bit to the surroundings.” ([35], p. 187).

Some observations about this result are in order. First, the principle is a consequence of the second law of thermodynamics, and therefore does not explain why the demon does not violate the second law in the first place. *Landauer’s principle does not resolve the paradox.* Indeed, a demon that is able to circumvent the second law would also be able to violate the Landauer limit and erase information without dissipation. Bennett himself has acknowledged the circularity of using Landauer’s principle to resolve the paradox. In his words, “[I]f the Demon is a thermodynamic system already governed by the Second Law, no further supposition about information and entropy is needed to save the Second Law. On the other hand, if the Demon is not assumed to obey the Second Law, no supposition about the entropy cost of information processing can save the Second Law from the Demon” ([48], p. 508). Second, in the Landauer argument there is assumed to be thermal contact between the system and its environment, that is, the system is implicitly assumed not to be isolated, contrary to Maxwell’s assumptions, but the analysis does not explain how this contact is achieved in the case of the traditional passive demon that has no contact with the gas particles. Our analysis shows that the system must be in thermal contact with its environment as well, and goes further by identifying and quantifying the “hidden” dissipation mechanism. Third, it is often suggested that the information erased by the process is somehow stored in the demon’s brain or some other memory storage device, which is external to the system, under the presumption that the demon must somehow record observations of the gas particles in order to decide whether or not to allow individual molecules to pass. While it is possible in principle to construct such a demon, the memory device then becomes part of the system and must be included in any analysis of it. In our treatment, there is no need for external memory storage. Rather, the information that is erased is stored *in the system itself*, serving as its own memory device—the reduction in $S_{sys}$ given by (Equation 22) *is* the erasure of information, at the rate $\dot{n}=-\frac{{\dot{S}}_{sys}}{k\ln2}$ bits per unit time. And finally, Landauer’s principle only accounts for enough information erasure to ensure that $\Delta S_{univ}\geq 0$, and so tells us no more about the process than the second law does. In particular, it allows for reversible operation of the demon, with ${\dot{S}}_{univ}$ arbitrarily close to 0. Our bound (Equation 23), on the other hand, is strictly positive for all nontrivial cases, putting more stringent limits on the system than Landauer’s principle, as discussed in the next section.

## 3. Discussion

The bound (Equation 23) is zero only in trivial cases ($\dot{K}=0$ and either $\dot{N}=0$ (no transport) or $N=N^{\prime }$ and $T=T^{\prime }$ (the sides are instantaneously in thermodynamic equilibrium with each other)). It is strictly positive otherwise, putting a greater restriction on the demon than the second law and indicating that the demon’s operation is thermodynamically irreversible. We can understand the nature of this irreversibility mechanically with the following “thought experiment on the thought experiment”: Imagine the system begins in some initial state, and the demon is turned on and allowed to operate for some time. Then at some instant, the velocity of every molecule in the system is reversed. In a reversible process, the system would thence evolve backwards along its original path and return to its original state. In Maxwell’s demon this cannot happen, because molecules that have been allowed to pass from the emitter to the receiver during the forward operation are prevented by the demon from returning to the emitter after their velocities are reversed. If one tries to patch this time-irreversibility by requiring that in addition to reversing particle velocities, the demon is also flipped in space, permitting particles to pass from the receiver to the emitter (that is, imposing parity reversal along with time reversal), one finds that the system still does not return to its initial state because now the demon allows some particles from the receiver that did not originally come from the emitter to pass into it along with the ones that did come from there. Hence, the irreversibility is a consequence of the spatial asymmetry of the demon’s sorting: particles on one side are subject to a different velocity filter than those on the other. Therefore, any Hamiltonian representing the demon cannot be separable into spatial and velocity parts, and it will have neither time nor time-parity reversibility.

It is sometimes suggested that irreversibility is a property particular to macroscale dynamics and is a consequence of the loss of detail in shifting from the microscale level of description to the macroscale. Such a model of irreversibility is motivated by the common assumption that all microscopic dynamical processes are reversible. Here, in contrast, we see a system whose dynamics are irreversible even at the microscale level of description that includes the positions and velocities of the individual particles. The entropy production at the macroscale is not due to lack of detail at that level, but to inherent irreversibility of the microscale dynamics. Systems like Maxwell’s demon with spatially-asymmetric, velocity-dependent dynamics at the microscale are indeed physically realizable. Examples include a Lorentz force diaphragm [49], asymmetrically laser-cooled optical traps [8,11,26,50], some carefully designed rectified electrical circuits [22,29], and so-called “non-Hermitian gyrotropic metasurfaces” [51]. Irreversibility can also be produced at the microscale by a Hamiltonian that is not velocity-dependent but is time-dependent [52].

We have here a clue to the connection between microscopic irreversibility and time’s arrow. Given the present velocities and positions of all particles in the system, the trajectory of any particle can be projected forward in time indefinitely. Similarly, the trajectory of any particle in the *emitter* can be projected backwards indefinitely, at least as far back as the demon has been operating. But for particles in the *receiver*, their backwards trajectory can only be projected as far back as a point when they emerged from the aperture, if they satisfied the demon’s velocity selection rule at that time. Before that, it is impossible to say whether they came from the emitter and were admitted through, or whether they came from the receiver and were reflected back into it. Hence the history of particles in the receiver can only be known probabilistically, because multiple possible microstates in the past map onto the present. Projecting forwards is clear, but projecting backwards is “hazy”. It is sometimes noted in reference to time’s arrow that “we can remember the past, but we cannot remember the future." In this case, the distinction between the past and future is not that the past can be remembered, but that it must be forgotten.

The bound (Equation 23) has been derived for the simplest Maxwell’s demon involving transmission in only one direction. The result is easily extended to demons allowing two-way transmission with different velocity selection rules each way, including “temperature only” demons that maintain the same pressure on both sides and “heat only” demons that transfer thermal energy with no net particle flux. The argument above scans exactly; only the evaluation of the averages in (Equation 4)–(Equation 7) would need to be modified to include both positive and negative $v_{z}$ and to reflect the different selection rules in each direction.

It is worth underscoring the effectiveness and potential utility of the main tool that has enabled this analysis, that is, the application of simultaneous mass, momentum, and energy conservation, leading to the identification of “hidden dissipation.” Although it has not previously been applied to Maxwell’s demon, the hidden dissipation effect is familiar from a class of mechanics problems involving momentum generation and energy transport in massive streams. These include the “conveyor belt problem,” in which sand is deposited onto a moving belt [53], a chain lifted off a table with constant speed, a gravity-driven siphon, and the “Mould effect” chain fountain [54]. A related problem in circuit theory is the constant-voltage charging of a capacitor, where the heat dissipated equals the work done on the capacitor regardless of the circuit resistance. In all these cases, as in Maxwell’s demon as we have seen above, an external force (generalized to include a potential difference in the case of the circuit) delivers both momentum (or current) and power to a system, the power delivered exceeds what can be contained in an organized form as macrosopic mechanical energy in the system, and the excess is dissipated. It is fundamentally a transport phenomenon, and since nonequilibrium thermodynamics is essentially concerned with transport, we should not be surprised to find it surfacing in a nonequilibrium thermodynamical situation like Maxwell’s demon. It seems likely that a similar approach will apply to other nonequilibrium thermodynamics problems as well, and that an accurate analysis of those problems will also reveal hidden dissipation. As a tool set for nonequilibrium thermodynamics, this framework has the additional benefit of dealing directly with rates of change, which are the natural variables of nonequilibrium thermodynamics but are by definition absent from equilibrium thermodynamics.

Previous attempts to explain the Maxwell’s demon paradox have been called “exorcisms,” whimsically playing on the image of the “demon” as a supernatural being and suggesting, consciously or unconsciously, the need to access other-worldly powers through arcane rituals to dispel it. Indeed, over more than a century, various “exorcists” have invoked salvation from the mysterious realms of Consciousness, Measurement, and most recently, Information. All these attempts have been fruitful in clarifying the connections between those heady domains and the physical world, but none has permanently succeeded in keeping the demon from haunting the Second Law house. The term “exorcism" does not aptly describe the explanation given in this paper, however, because it does not draw on new or exotic physics, but remains within the earthly plane of Conservation of Particles, Momentum, and Energy. To extend the metaphor, we have, upon inspecting the house, discovered that what has been unsettling us for 150 years, making eerie noises and inexplicably piling the cereal up in one corner of the kitchen cupboards, was not a demon or ghost after all, but just a family of squirrels who have taken up residence in the walls and have been doing their work and producing dissipation there all along. Rather than Maxwell’s demon, it appears we have Maxwell’s vermin. As for the currently entrenched explanation from information theory, we have shown it to be logically circular and to miss the essential irreversibility that defines the problem, and anyway, to quote Laplace’s legendary retort to Napoleon, “Je n’avais pas besoin de cette hypothèse-là” [55].

## Figures and Tables

**Figure 1 entropy-24-00093-f001:**
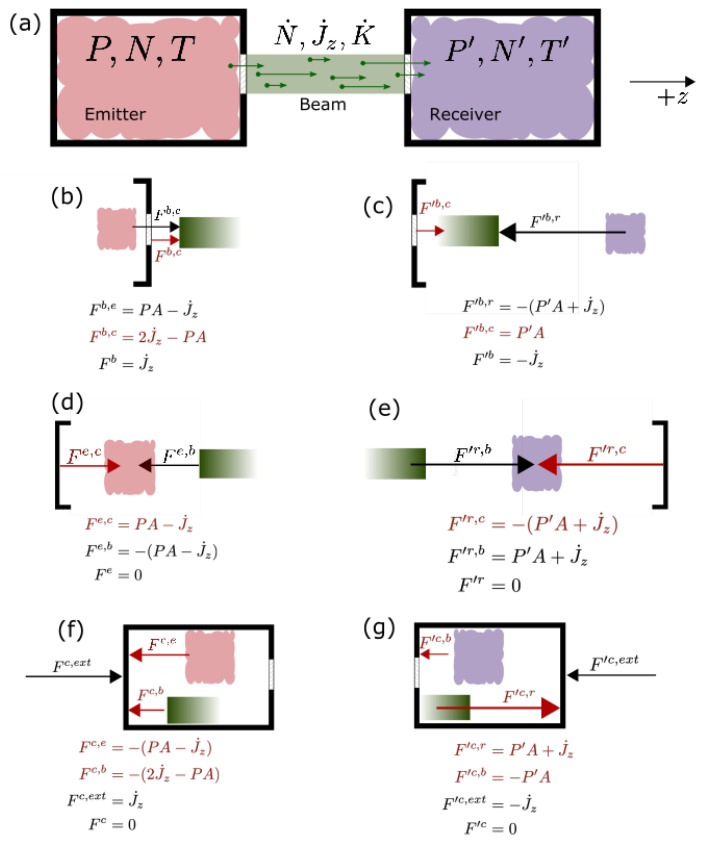
Schematic representation of Maxwell’s demon, showing (**a**) the system of gases, the container (including both chambers and the sorting mechanism), and the beam of particles, (**b**,**c**) forces on the beam at the emitter and receiver ends, (**d**,**e**) forces on the gases at the emitting and receiving ends, and (**f**,**g**) forces on the chamber at the emitting and receiving ends. Internal forces exerted by or on the container are labeled in red. Each subfigure lists the component forces and the net force on that element.

**Figure 2 entropy-24-00093-f002:**
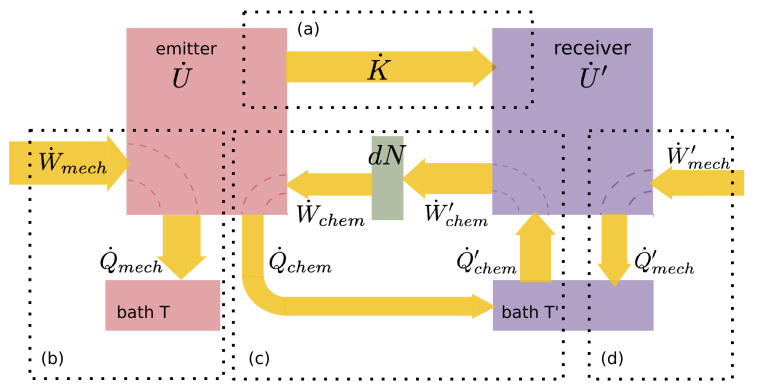
Schematic representation of energy flow. Work is positive if done on a subsystem; heat is positive if leaving a subsystem. Processes are delineated with dotted boxes: (**a**) Kinetic energy transport from the emitter to the receiver. (**b**,**d**) Mechanical work by the container on the emitter and receiver, dissipated as heat to the respective heat bath. (**c**) Thermal chemical work in transferring particles, which do work at the rate ${\dot{W}}_{chem}\left(>0\right)$ on the emitter and at the rate ${\dot{W}}_{chem}^{\prime }\left(<0\right)$ on the receiver, because the particle flux is from emitter to receiver. The net effect of cycle (**c**) is to convert heat from the $T^{\prime }$ reservoir into work at the rate $-({\dot{Q}}_{chem}+{\dot{Q}}_{chem}^{\prime })$, or equivalently, $-({\dot{W}}_{chem}+{\dot{W}}_{chem}^{\prime })$. This is positive if $T<T^{\prime }$ (as shown).

## Data Availability

No new data were created or analyzed in this study. Data sharing is not applicable to this article.

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
