# Peer review of "Hidden Dissipation and Irreversibility in Maxwell’s Demon"

_entropy, 2022, doi:10.3390/e24010093_

Round 1

Reviewer 1 Report

The claim is strong, but the derivation should be clearer. There are several points where I trip when reading the manuscript. Therefore, I would like to ask the author to clarify the following points:

1) In equations 1, 2, and 3, I do not understand why there is no energy of the Maxwell demon although the classical solution of the paradox actually makes the demon part of the system. 

2) In equations 4 and 9 (and at other places) the equilibrium thermodynamic quantities like pressure and temperature appear. The equilibrium quantities are well defined if the distribution function is Maxwellian. But the filter actually alters the velocity distribution, making it non-Maxwellian. Therefore, I suggest carrying out the analysis for instance completely on the kinetic level of description so that the validity is clear even if the demon alters the distribution function. The local equilibrium quantities become accessible only after the thermalization (restoring the Maxwellian distribution), see e.g. https://doi.org/10.1515/9783110350951

3) I do not understand why equation 14 should be valid. The principle of minimum entropy production, if the author wanted to use it, does not work generally. Actually, it only works I the case of linear irreversible thermodynamics with constant phenomenological coefficients, see e.g. https://books.google.cz/books/about/Understanding_Non_equilibrium_Thermodyna.html?id=oN9PprnxKncC&source=kp_book_description&redir_esc=y

In summary, the claim of the manuscript is definitely interesting, but there are several places in the calculations that are difficult to follow and thus hard to judge, at least for me. For instance, relying purely on kinetic theory without invoking any other principles could be a remedy. It is also unclear to me to what extent the demon is part of the system.

Reviewer 2 Report

In this paper, the author tries to explain the irreversibility in Maxwell's demon starting from kinetic theory. There are several problems with the manuscript. For example:
- the assumptions made in this paper to derive the main results, namely that the total energy of the system is constant and that the separate parts in Eq. 3 are zero, are generally wrong, and it is not clear to me why these assumptions would be better than assuming the second law.
- the main derivations seem to be wrong. For example, the average velocity of particles in the beam, as derived in appendix A, does not agree with the existing equations for effusing particles (see e.g. https://en.wikipedia.org/wiki/Effusion)
- the paper is generally very hard to read. For example, the definition of the rate of change of momentum is not clear to me. Is this the rate at which momentum is transferred from the left to the right reservoir? And if so, why can one say that dW=vdJ?

For these reasons I think that this manuscript will only add confusion and therefore I cannot recommend it for publication.

Reviewer 3 Report

This is an interesting and challenging paper. It presents a novel idea that is worth publishing in order to continue the discussion of this demon. Hopefully, the "demon status" will be finally removed.

Round 2

Reviewer 1 Report

Thank you for implementing the changes. 

Let me please comment a bit more on a few issues:

1) I see that you take the demon out of the system. But the demon is still interacting with the gas by letting some molecules pass. Although it might be thought of as not adding any energy into the system, it acts as an entropy pump. And since you violate the equilibrium distribution, you no longer have the direct link between entropy and heat fluxes. So, it seems that you have an entropic pump, which would need some external power to compensate for its work (e.g. heating up). What would happen if you considered such external source?

2) Regarding the question of microscopic reversibility vs macroscopic irreversibility, that you discuss at the end of your manuscript, I still can't see why forgetting the past would cause the irreversibility. Maybe in your example could be seen that way, but generally it is the forgetting of the present details (for instance https://link.springer.com/article/10.1007/s10955-020-02563-7) that causes the emergence of irreversibility, while the state of the system could be defined at a given time instant completely (in theory). In other words, a bunch of classical particles is completely described by their positions and momenta at the moment, but by their history. If we disregard some details about the present state, we observe irreversible behavior. In your example, I'd say that you add an entropy pump, which creates the asymmetry. How is your forgetting history compatible with the approach based on forgetting details at the present?

Reviewer 2 Report

The presentation of the manuscript has improved a lot and the author managed to answer several of my concerns with the previous version of the manuscript. I do still have some major concerns about the derivation of the main results which make it impossible for me to accept the current version of the manuscript, but I would be happy to look at a revised version if the author manages to make the necessary changes/clarifications.

My most important concern is the derivation of Eqs. 8-11. I have problems following this derivation, and the results seem contradictory to me. Consider for example the situation where there is no demon but the density of the gas in the receiver is much lower than the density of the gas in the emitter. In this case, the particle and momentum fluxes, and the pressure could essentially be the same. However, as there is no demon, Eq. 9 should go to 0. How would this happen? A similar problem exists with the introduction, where the author says that momentum transport needs to be balanced by an external force. Why would this be necessary for a system with a demon and not when there is a momentum flux purely due to a pressure difference?

Another smaller concern is that the author uses the ideal gas law to derive Eq. 11. This only holds if the gas is in equilibrium, which technically is not the case. Does the fact that one can use the ideal gas law follow from the weak demon assumption?

Round 3

Reviewer 2 Report

I have to admit that I still have problems following the derivation of Eqs. 8-12. This might partially be due to my different scientific background (stochastic systems), but it might help to give a more detailed explanation on how the demon operates. Also, the presence of a mechanical work (Eq. 11) is still confusing to me. I do not see how any mechanical work would be present in the original formulation of Maxwell's demon.

For this reason, I feel like I cannot give a conclusive recommendation on this manuscript.